# Quantification of Visual Attention by Using Eye-Tracking Technology for Soundscape Assessment Through Physiological Response

**DOI:** 10.3390/ijerph21111478

**Published:** 2024-11-07

**Authors:** Hyun In Jo, Jin Yong Jeon

**Affiliations:** 1Department of Architectural Engineering, Hanyang University, Seoul 04763, Republic of Korea; best2012@hanyang.ac.kr; 2Department of Medical and Digital Engineering, Hanyang University, Seoul 04763, Republic of Korea

**Keywords:** eye tracking, visual attention, physiological measurement, soundscape quality, psychological restoration

## Abstract

Because soundscapes affect human health and comfort, methodologies for evaluating them through physiological responses have attracted considerable attention. In this study, we proposed a novel method for evaluating visual attention by using eye-tracking technology to objectively assess soundscape perception. The study incorporated questionnaire surveys and physiological measurements focusing on visual attention responses. Results from the questionnaire indicated that perceptions of vehicles and the sky were 6% and 26% more sensitive, respectively, whereas perceptions of vegetation, based on physiological responses, were approximately 3% to 50% more sensitive. The soundscape quality prediction model indicates that the proposed methodology can complement conventional questionnaire-based models and provide a nuanced interpretation of eventfulness relationships. Additionally, the visual attention quantification technique enhanced the restoration responses of questionnaire-based methods by 1–2%. The results of this study are significant because the study introduces a novel methodology for quantifying visual attention, which can be used as a supplementary tool for physiological responses in soundscape research. The proposed method can uncover novel mechanisms of human perception of soundscapes that may not be captured by questionnaires, providing insights for future research in soundscape evaluation through physiological measurements.

## 1. Introduction

Reducing noise problems is crucial for preserving the health of city dwellers and realizing sustainable urban environments [1,2]. Countries around the world have conducted various studies and prepared legal systems for regulating and managing urban noise [3,4]. However, studies have revealed that reductions in noise levels do not necessarily guarantee the satisfaction of city dwellers. Consequently, a novel concept of soundscape has emerged [5,6]. In the revised ISO 12913-1 [7], a soundscape is defined as an “acoustic environment as perceived, experienced, and/or understood by a person or people”. In contrast to previous concepts in the noise field, the soundscape concept is focused on “wanted sounds”, such as birdsongs or water sounds, instead of noise-based “unwanted sounds”. Evaluating soundscapes involves examining the relationship between a descriptor, which quantifies the human perception of an acoustic environment, and an indicator, which predicts the descriptor [8]. Soundscape evaluation methods can be classified into soundwalks, laboratory experiments, narrative interviews, and behavioral observation; evaluation tools can be classified into questionnaires, interview protocols, physiological measurements, and observation protocols [9,10,11,12]. Questionnaires are widely used because of the ease of generalization and statistical analysis, and ISO 12913-3 [13] suggests detailed manners to analyze questionnaire-based evaluations. However, evaluation methods using such closed questionnaires force subjects to focus on specific items by limiting in advance the response ranges and can cause bias such as acquiescence bias or demand characteristics [14]. To overcome this problem, interviews can be used. However, their analysis is more complex than that of questionnaires, and the interpretations of the same results can vary depending on the sociocultural backgrounds of researchers [15,16].

Various studies have been conducted to determine tools that can objectively evaluate human perceptual responses in the soundscape field [8,11]. However, existing methodologies are typically complex evaluations and designed for small groups of people. To use soundscape research for policy purposes, we proposed a simple and objective evaluation tool suitable for large groups of subjects [11]. A possible solution is to obtain physiological measurements to objectively evaluate the subjects’ perceptions of their soundscape experiences. These measurements include the heart rate, R-wave amplitude, heart-rate variability, electroencephalography alpha reactivity, electroencephalography beta reactivity, forehead electromyography, eye blink frequency, respiratory frequency, respiratory rate, respiratory depth, skin conductance, and skin temperature [17,18,19,20,21,22]. Numerous studies on the stress or restoration responses of subjects to nature- and noise-related sounds have been conducted for pleasant soundscape realization. The aforementioned indices are related to the affective or physiological restoration responses of humans and are intended primarily to evaluate the effects of experiences on psycho-physiological responses.

Considering the surrounding context of a soundscape is crucial and necessitates a multisensory study that involves various senses other than hearing [6,7,8]. The concept of soundscape is derived from the concept of landscape, and a large part of the five human senses relies on sight and hearing. Numerous studies have been performed on the effect of audio–visual interaction on soundscape perception [16,23,24,25,26]. Researchers study audio–visual interaction because soundscapes are related to the context, of which the visual environment is an important part. Studies have focused on changes in preferences (satisfaction, comfort, pleasantness, etc.) for soundscapes and landscapes according to whether audio and/or visual information is provided, and based on these studies, urban planning design and management strategies have been developed. For example, Nauener Platz in Germany won the Soundscape Award from the European Environment Agency and the UK Noise Abatement Society for its “Audio Island,” which was created by remodeling the park and incorporating various sound elements for citizens to enjoy [27]. Brighton and Hove, UK, developed ambient audio technology to improve crowd behavior and reduce police intervention [28]. As another example, at the central train station in Sheffield, England, visual and auditory elements were integrated through soundproof walls, symbolizing hydraulic elements and the steel industry, to create a pleasant urban environment [29]. As presented in these examples, the concept of soundscape has sufficient practical value in creating pleasant spaces for urban residents by integrating visual and auditory elements at the urban planning stage. To consider the effect of landscapes on soundscape assessment, the physical characteristics of visual environments are quantified as red–green–blue ratios, landscape spatial patterns, landscape compositions, etc. [30,31], or questionnaire-based research methods for evaluating landscape perception are used [25,32].

Visual attention indicates the capability of a person to selectively notice visual stimuli in the visual environment and remember only what is required [33,34]. Visual attention has become a critical evaluation index in the soundscape field. Visual blocking designs for noise [35] or moving objects change human visual attention, which affects soundscape perception [36]. Jo and Jeon [37] verified that human-related sound source identification becomes prominent as human visual attention increases. Eye tracking is the most widely used method for quantifying visual attention according to physiological responses [38,39,40]. Liu et al. [38] demonstrated that eye-tracking technology could replace visual aesthetic quality and tranquility rating evaluation in space. Ren and Kang [36] found that artificial sounds enlarge visual attention areas in tranquility evaluation and that natural sounds enlarge visual attention areas for natural landscapes. However, few studies on the soundscape field have quantified eye movement. Even in the aforementioned studies, in which visual attention responses were examined, two-dimensional (2D) stationary pictures were used, and time-series change characteristics, such as events and activities in real space, were not considered.

As aforementioned, an ongoing necessity exists for objective evaluation tools for soundscapes, and studies on the development of these tools are continuously conducted. However, limited studies have investigated applying objective evaluation tools focusing on the visual sense of soundscapes. Furthermore, soundscape studies on the perception of visual elements have been primarily questionnaire-based, and few studies have been physiological measurement-based. For academic, industry, and society officials to be able to manage efficiently and continuously the urban sound environment, a methodology that replaces subjective evaluations with objective methods is essential (particularly because subjective evaluation methods are time-consuming and expensive). Therefore, we proposed and verified a novel methodology to objectively quantify visual attention based on human physiological responses in soundscape studies. We proposed quantifying visual attention in dynamic environments using eye-tracking technology and deep-learning algorithms. The effectiveness of the proposed method was verified through comparison with conventional questionnaire-based evaluation methods.

Accordingly, the research questions for this study are as follows:(1)Is there a novel methodology to quantify visual attention as static and dynamic objects based on eye movement measurements?(2)Do the results of visual attention evaluation using the conventional questionnaire method and those using the physiological measurement method proposed in this study differ?(3)Is the visual attention quantification method proposed in this study appropriate for evaluating the overall quality and restoration of soundscapes?

## 2. Methods

### 2.1. Quantification Methods for Visual Attention

In this study, eye-tracking technology was used to quantify the visual attention to a soundscape. Eye-tracking technology, which is a scientific measurement tool for tracking eye movement by perceiving pupil centers and corneal reflexes, quantifies eye movement by using near-infrared lighting and cameras [41]. Indices associated with fixation and object detection were selected among various indices for quantifying eye movement [42]. The indices are defined as follows. Fixation refers to eyes remaining within 2° of a threshold of dispersion for ≥300 ms, whereas object detection refers to eyes remaining focused on an area of interest (AOI) for a certain period [41]. The reason for designating an AOI in the process of quantifying object detection is as follows. Visual attention data are generally expressed in the form of displayed eye movement when stimulating images are provided to subjects. AOIs are set for researchers to identify information when the eyes of subjects remain focused on specific parts or objects and determine the overall eye movement of the subjects. As aforementioned, because eye locations are represented as coordinate values over two-dimensional (2D) images, the AOI of the corresponding coordinates should be identified prior to an experiment. First, the visual elements of the sites evaluated in this study are categorized into static and dynamic factors. The static objects are background elements and are classified as buildings, vegetation, water space, and sky. The dynamic factors are foreground elements and are divided into vehicles and people. The AOIs for the static factors should be set only once because the static factors are fixed elements and change little over time. Therefore, using Adobe Photoshop (v. 2025), various colors are assigned to different factors to form colored layers (Figure 1c). For the dynamic factors, the AOIs should be periodically modified because they move continuously. However, manually dividing layers for each frame of a video is not feasible. To effectively perform this process, the You Only Look Once (YOLO) v4 object-detection algorithm is used [43]. Moving vehicles and people are continuously perceived, and color layers are assigned to the perceived factors (Figure 1d). YOLO is a deep-learning technique that simultaneously “predicts the locations and sizes of objects” and “classifies the classes of objects.” YOLO v4 was used in this study because it accurately detects small objects and is an optimized model that can be trained and tested with a single graphics processing unit. Among various object detection models, YOLO v4 provides high mean average precision and frames per second performance simultaneously [43]. Additionally, three researchers re-verified specific intervals where moving objects (dynamic factors) appeared in the YOLO v4 object detection results to minimize errors. The operation concept of the eye-tracking process is illustrated in Figure 2. The subjects are provided only with the original scenes of the three-dimensional (3D) audio–visual environments. The dynamic and static factors divided by the AOIs are hidden from the subjects and are used only in the calculation process. The measurement output for eye movement is recorded as fixation points on a plane by converting 3D coordinates into 2D coordinates.

In this study, the eye tracker built into VIVE Pro EYE was used to quantify visual attention [44]. The Gaza data output frequency (binocular) of the corresponding hardware was 120 Hz, and the trackable was <110° when the two eyes were combined. For a single eye, the tracking accuracy (offset between recorded eye points and real locations) was 0.5°–1.1° for a FOV of <20°. For the eye-tracking interface, the HTC SRanipal SDK was used in a Unity 3D engine environment. We calculated the attention indices of the two eyes, number of fixations, and time of fixation for the six factors classified here, namely, vehicles, people, buildings, vegetation, water space, and sky.

**Figure 1 ijerph-21-01478-f001:**
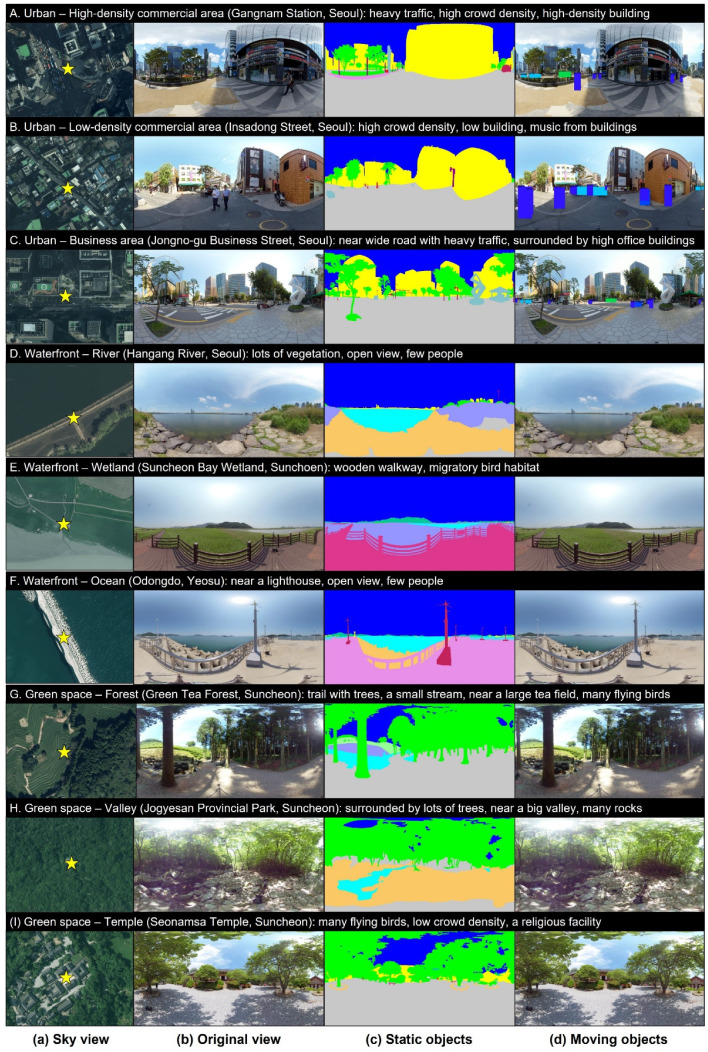
Nine evaluation sites: (**a**) sky view, (**b**) stitched monoscopic 360° original view, (**c**) color layers of static factors [45]; and (**d**) moving-object detection; A–C urban areas, D–F waterfront areas, and G–I green areas.

**Figure 2 ijerph-21-01478-f002:**
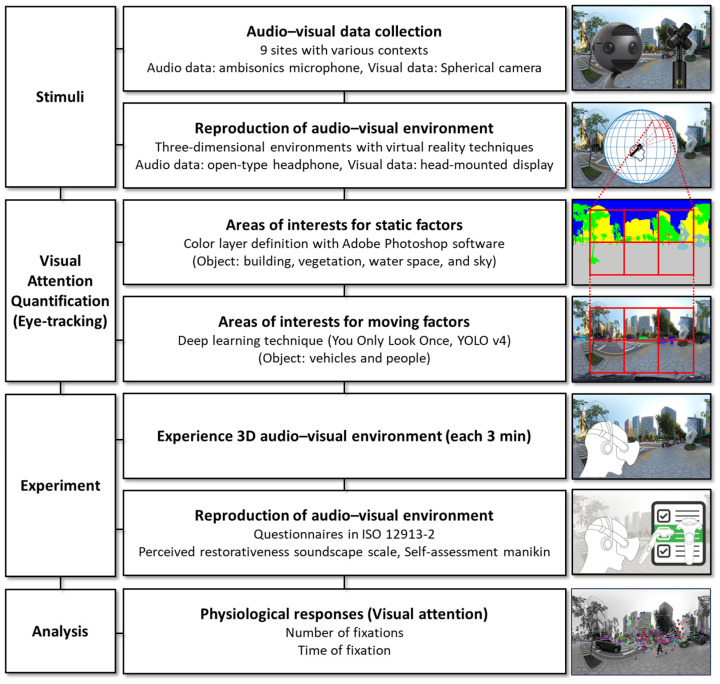
Study procedure and eye-tracking technique concepts for the detection of static and dynamic objects [9].

### 2.2. Stimuli

#### 2.2.1. Audio–Visual Data Collection

As depicted in Figure 1, nine sites were selected as environments for visual attention evaluation. To encompass various contexts, these nine sites included urban, waterfront, and green spaces, all of which were located in Korea. The urban spaces included a high-density commercial district, low-density commercial district, and business area and were characterized by buildings of various scales and traffic and people with high moving rates. In particular, the area near Gangnam Station in Seoul, designated as a high-density commercial area, is characterized by a very dense concentration of commercial and office facilities, high population mobility, and vibrancy. Most high-rise buildings are densely packed with glass or metal materials, and many wide roads, large billboards, and digital signage exist, providing a modern and sophisticated urban landscape. By contrast, Insadong Street, designated as a low-density commercial area, is a region in which conventional culture and tourism-centered businesses are developed. The model is characterized by hanok buildings and low-rise shops, creating a tranquil and leisurely atmosphere centered around pedestrians. The narrow alleyways and conventional-style exteriors shape the overall landscape, highlighting conventional charm in contrast to Gangnam Station. We selected a point where the visual characteristics of these two commercial areas could be well compared and conducted the study. The waterfront spaces included a river, a wetland, and an ocean and were characterized by water sounds and more open landscapes than the other sites. The green spaces included a forest, valley, and temple and were characterized by birdsongs and a prevalence of trees compared with other sites.

Audio–visual data were collected at the nine sites to construct the evaluation environments in a laboratory. The measurements were performed during the daytime (10 AM–2 PM) in May 2020. High-definition (8K) visual information was recorded using a six-channel spherical camera (Insta360 Pro, Shenzhen, China). Audio information was recorded in the A-format first-order Ambisonic format by connecting a four-channel Ambisonic microphone (SoundField SPS200, SoundField, Maroochydore, Australia) to a portable sound recorder (MixPre-6, Sound Devices, Parramatta, Australia). The audio information was synchronously recorded using a correctional microphone (half-inch microphone, AE 46, GRAS, Skovlytoften, Denmark) to adjust sound pressures to the same level. The measurement height of the camera was fixed at 1.6 m, that is, the average human eye level, and data were collected for 3 min. The measurement height of the microphones was fixed at 1.5 m. Because simultaneously setting all the measuring instruments to the same height is difficult, visual height was given priority.

As presented in Table 1, the acoustic characteristics of the selected sites were as follows: *L_Aeq_* ranged from 44.3 to 75.9 dB, *L_Ceq-Aeq_* ranged from 6.3 to 27.2 dB, and *L_A5-A95_* ranged from 5.2 to 21.5 dB. Thus, each of the parameters exhibited a max–min difference >10 dB, indicating that the stimuli covered sufficient variations in the typical outdoor sound setting. The visual parameter was quantified as the color ratio and is presented in Table 1 [45,46]. After converting the 3D captured video into a 2D image, the segments for the layer distinguished in Figure 1c were classified into vegetation (trees, grass, etc.), water features (fountains, rivers, etc.), sky, and artificial elements (vehicles, roads, buildings, etc.). Through the image color summarizer [47], the proportion [%] of each segment occupying the entire 2D image was calculated and named Green, Water, Sky, and Grey. Thus, the Grey [%] was the highest at 62–73% in the urban area, the Sky [%] accounted for 45–47% in the waterfront area, and the Green [%] accounted for 37–41% in the green space.

#### 2.2.2. Reproduction of Audio–Visual Environment Through Virtual-Reality Techniques

A verified audio–visual environment reproduction technique was used to construct realistic test environments in a laboratory environment [16,48]. In a previous study [48], the soundscape evaluation results between a VR lab environment implemented using various acoustic reproduction methods and a real environment were compared. In particular, when providing binaural sound-reflecting head movements based on first-order ambisonics in a VR environment, the overall soundscape quality evaluation results were statistically significantly identical to the actual in situ evaluation results (*p*-value < 0.05). This phenomenon demonstrates that soundscapes can be evaluated in a realistic and reliable manner even in a VR environment. In this study, we used the same VR evaluation methodology to conduct assessments under conditions similar to the real environment, even within the laboratory. First, videos collected using the six-channel camera were edited into a single 360° video using stitching software (Insta360 Pro Stitcher, v3.0.0), and the 360° video was displayed to the subjects by using a head-mounted display (HMD, VIVE Pro EYE) with a built-in eye tracker. Next, the Ambisonic soundtrack recorded in four channels was transformed from A-format into B-format using the Ambisonics Plugin (Sound Devices) in MixPre-6. Next, the Ambisonic sound was downmixed into a binaural track and reproduced to the subjects through open-type headphones (HD 650, Sennheiser, Wedemark, Germany). The downmixed sound was recorded with a head and torso simulator (Type 4100, Brüel & Kjær, Darmstadt, Germany) to adjust the sound pressure level to the same *L_Aeq_* value measured with the correctional microphone and was edited using the amplitude function of the Adobe Audition software (v. 1.5, Adobe).

### 2.3. Experimental Design

#### 2.3.1. Participants

Prior to experiments, the required number of samples was computed using the G*Power software (v. 3.1.9.4). The results revealed that at least 42 subjects were required for verification with a statistical effect size of 0.5, verification power of 0.8, and significance level of 0.05. We recruited students attending universities located in Seoul to reduce the response errors between subjects. A total of 60 subjects were recruited for this study. All the subjects had normal eyesight (corrected eyesight ≥ 1.0). Because of a reliability problem caused by ophthalmodonesis in 10–20% of ordinary people, error rates were considered [42]. Moreover, in addition to decreased vision and eye movement with age, non-normal vision can cause errors in the eye-tracking process because of the refraction of light by glasses or lenses. Therefore, only normal-vision subjects in their 20s were recruited for and evaluated in this study. Before the test, the audiometric thresholds of the subjects were evaluated using an audiometer (Rion AA-77), and it was confirmed that all subjects had normal hearing levels. The number of males was equal to the number of females, and the subjects were 21–30 years old (mean age of 24.3 years and standard deviation of 2.4 years).

#### 2.3.2. Questionnaires

The questionnaires for evaluating the audio–visual perception of stimulation comprised three parts composed of the following ISO 12913-2 [9]. When a questionnaire translated into Korean was provided, the original English text of the ISO standard [9] was included to preserve the meaning of the original text. The first part contained items associated with visual element identification for comparison with the eye-tracking technology. The subjects were asked to evaluate the degrees to which vehicles, buildings, people, vegetation, water, and sky were perceived on a five-point Likert scale. The question was “To what extent do you presently see the following six types of visual elements?”. The subjects responded on a scale from 1 (not at all) to 5 (dominates completely) regarding how well each visual element (e.g., vehicle, vegetation, building, people, water space, and sky) was recognized. For example, if the average response for “vegetation” is 4 points, then most respondents clearly recognized that element. By contrast, if the average response for “vehicle” is 2 points, then most participants recognized vegetation less. In the second part, soundscape perception was examined. First, the subjects were asked to evaluate the degrees to which traffic noise, other noises, human sounds, and natural sounds were perceived on a five-point Likert scale. The question was “To what extent do you presently hear the following four types of sounds?” Next, the subjects were asked to rate the degree to which they agreed with eight semantic expression words (pleasant, vibrant, eventful, annoying, chaotic, calm, monotonous, and uneventful) for evaluating the perceived affective quality of the soundscape and two factors (overall impression and appropriateness) for evaluating the overall perception on a five-point Likert scale. In accordance with the recommendations of ISO 12913-2 Annex C [9], all scales were presented in the same direction regardless of positive or negative values. For example, a positive scale like “pleasant” indicates a more positive evaluation with a higher score, and a negative scale like “annoying” is designed so that a higher score indicates a stronger negative evaluation. This is to maintain the consistency of the scale and to reduce confusion when respondents answer the survey. The third part included items for examining affective restoration responses through audio–visual environment experiences. We constructed a perceived restorativeness soundscape scale (PRSS) [49] and divided the restoration effect of the soundscape into six domains, namely, fascination, being-away-to, being-away-from, compatibility, extent (coherence), and extent (scope). We subsequently evaluated the six domains according to the attention restoration theory and self-assessment manikin (SAM) [50], dividing the affective responses to stimuli into two domains—valence and arousal—and examining the two domains. Seven- and nine-point scales were used for the valence and arousal, respectively.

#### 2.3.3. Procedure

The visual attention evaluation procedure of this study was as follows. First, the subjects visiting the laboratory were provided with a simple explanation of the questionnaire items, including the soundscape concept. The subjects were informed that their eye movements would be tracked but were not informed of the tracking time to avoid bias. After the explanation, the subjects wore the provided HMDs and headphones. Prior to the evaluation, each subject underwent a calibration process for their eye movement to increase the precision and accuracy of the eye-movement measurements. During the calibration, the subjects repeatedly gazed at points displayed at random positions on the screen [51]. When the evaluation preparation was completed, the subjects rested for 1 min, after which they were provided with nine stimuli in random sequences. The subjects filled out the questionnaires regarding the same stimulation after a 3 min experience with the audio–visual stimuli. In this study, each stimulus was provided for 3 min to minimize the fatigue of participants because of the experiment by referring to the minimum evaluation time recommendations in ISO 12913-2 [9] for obtaining reliable soundscape measurement results.

A critical consideration was that the subjects recognized that the stimuli would be provided one more time for the survey response after the first exposure. This awareness among the subjects could have caused errors in visual attention. We could not ensure that the subjects did not take the initiative to look at various visual elements when they were provided again with the stimulus because unlike the heart rate, blood pressure, and other physiological indicators, eye movement can be subjectively controlled. Therefore, we quantified visual attention using only the response when the subject was first exposed to the stimuli.

Every questionnaire was filled out using a controller within the virtual-reality environment. Consequently, 540 results (9 sites × 60 participants) were collected for each eye-movement quantification index and questionnaire item.

### 2.4. Data Analysis

In this study, the following analysis was performed using SPSS Statistics (v. 25, IBM). First, the normality (Shapiro–Wilk and Kolmogorov–Smirnov tests) and homoscedasticity (Levene) of all the response data were tested. Thus, the normality requirements were satisfied, and parametric statistics were calculated. The relationship between the visual attention evaluated with the eye-tracking technology and the questionnaire responses was examined using the Pearson correlation. Next, the effects of attention to the visual factors on soundscape perception and affective restoration were examined, and multiple linear regression analyses were performed to compare the questionnaire and physiological response results.

## 3. Results

### 3.1. Visual Attention Response

#### 3.1.1. Physiological Response from Eye-Tracking Technology

The evaluation results for the attention quantification method based on the eye-tracking technology and the proposed object-detection algorithm are presented in Figure 3. Figure 3a illustrates the number of fixations, which represents the number of times the eyes remained on each AOI. The visual attention responses varied according to the context of each site. The degrees of visual attention to the artificial elements (buildings and vehicles) and humans were high for the urban environments, whereas the degrees of attention to the natural elements (vegetation, water space, and sky) were high for the natural environments. Specifically, the eyes focused mostly on the water space and sky in the waterfront areas, whereas the degree of visual attention to the vegetation was high for the green areas. For parts of the green areas (sites G and H), the degree of attention to the water space was high. A similar tendency is observed in Figure 3c, which presents the relative ratio of each AOI obtained by converting the number of fixations into a percentage (%). The overall accumulation of the number of fixations varied considerably. A high frequency of eye movement (≥175 times) was observed for the urban environments, whereas a low distribution (≤141 times) was observed for the natural environments. In terms of the number of fixations according to the spatial context, a large, accumulated difference was observed between the urban and natural environments. By contrast, as presented in Figure 3b, which presents the duration of fixation of the eyes, the relative differences between the urban and natural environments were reduced for all sites except site E. The overall accumulated time of eye fixation was distributed across the range of approximately 116–140 s for the urban environments and 90–104 s for the natural environments. A similar tendency was observed in the relative distribution of the duration of fixation on the AOI of each space (see Figure 3d). Thus, eye movements change considerably in urban environments, where the multitude of events and human activities that occur generate a complex context, leading to visual attention being widely distributed among various visual elements. By contrast, in natural environments, eye movements are more limited because the eyes tend to focus on natural features for longer periods, resulting in concentrated visual attention on these features.

#### 3.1.2. Psychological Responses to Questionnaire

Figure 3e depicts the identification results from the questionnaires examining various visual elements constituting the space. The dominance percentage, representing the ratio of people who evaluated each visual element and gave three or more points on the five-point Likert scale, was calculated to examine the visual attention responses. A Pearson similarity (correlation) analysis was performed to identify the similarity between the visual element identification results and physiological response results. The similarities between the number of fixations and the visual element identification were 0.56 for vehicles, 0.80 for buildings, 0.76 for people, 0.10 for vegetation, 0.70 for water, and 0.42 for the sky. The similarities between the duration of fixation and the visual element identification were 0.57 for vehicles, 0.83 for buildings, 0.77 for people, 0.25 for vegetation, 0.70 for water, and 0.49 for the sky. For the vehicles, sky, and vegetation, differences were observed between the questionnaires and the physiological responses.

To identify the causes of such differences, the physiological responses and the visual element identification results were quantitatively compared for each evaluation point. Several areas with low perception ratios in the physiological responses had high values in the corresponding visual element identification. Specifically, for the urban environments, vehicle and sky perceptions were 20–25% and 15–16%, respectively, which were higher than the corresponding physiological response results in Figure 3a,d. The natural environments also exhibited notable trends. Compared with the questionnaire results regarding the perception of water features in the waterfront areas (29–42%), the physiological responses to the same features were higher (33–55%). Moreover, the visual perception of vegetation was considerably higher (15–28%) compared with the corresponding physiological response (1–15%). In contrast to the low 30–33% exhibited by the questionnaire results regarding the perception of vegetation in green spaces, a high distribution of 54–90% was exhibited by the corresponding physiological response results. The degree of attention to the sky was high (26–42%) in the questionnaire results but was not notable in the physiological responses. The visual attention responses obtained when the subjects were allowed free experiences differed considerably from the visual attention responses obtained through the questionnaires designed by the researchers.

#### 3.1.3. Relationship Between Physical Characteristics and Visual Response

A Pearson correlation analysis was conducted between physical characteristics and visual response to examine the relationship between objective environment difference and visual response, and the results are summarized in Table 2. Generally, significant differences were not observed between eye-tracking and survey methods, and their relationship with physical characteristics was similar. First, spaces with many vegetation-related elements visually had a positive correlation with visual response to vegetation, and spaces with many water-related elements, such as fountains or rivers, had a positive correlation with visual response to water space. Environments with a high distribution of artificial elements, resulting in a high gray ratio, revealed a strong positive correlation with vehicles, buildings, and people. This phenomenon indicates that in densely built-up areas such as commercial or business districts, a high flow of people exists, and consequently, the visual perception of people also increases. Examining the relationship between acoustic parameters and visual response revealed that areas with high *L*_Aeq_ and *L*_A5-A95_ had a positive correlation with vehicles, buildings, and people. Furthermore, *L*_Aeq_ and nature-related visual response had a negative correlation, which revealed that the visual response to vegetation, water space, and sky tended to be highly evaluated in quiet places.

### 3.2. Soundscape Perception

#### 3.2.1. Soundscape Assessment

Figure 4 depicts the soundscape perception results obtained from the questionnaires and the assessment method proposed in ISO 12913-2 [9]. As indicated by the sound source perception results in Figure 4a, the perceptions of traffic and other noise were ≥3.70 and ≥3.43, respectively, in the urban environments, and the perception of human sounds was at least 3.60 because numerous people passed by in the urban spaces. In the natural environments, the natural sounds were dominant (at least 3.43) in both the waterfront areas and green spaces. The perception of the other noise-based sound sources was low (at most 1.70). The perception of human sounds was at most 2.7 in the green spaces, that is, lower than 3—the median on the Likert scale.

The results for the perceived affective quality of the soundscape are presented in Figure 4b. The pleasantness and eventfulness were calculated as suggested in ISO 12913-3 [13] according to the response results for the eight semantic expression words. The pleasantness was calculated as (*pleasant* − *annoying*) + cos 45° × (*calm* − *chaotic*) + cos 45° × (*vibrant* − *monotonous*). The eventfulness was calculated as (*eventful* − *uneventful*) + cos 45° × (*chaotic* − *calm*) + cos 45° × (*vibrant* − *monotonous*). In the urban environments, the pleasantness was lower than 0; however, the eventfulness was high (at least 3.77). This phenomenon could be attributed to the various moving objects, such as people and vehicles, in the urban environments, as inferred from the perception results for the audio–visual elements described previously. In the natural environment, the waterfront areas and the green spaces differed slightly. The pleasantness was in the range of 2.06–2.55 for the waterfront areas, whereas a rather high distribution of 2.39–3.37 was obtained for the green spaces. For the eventfulness, because the waterfront area provided a calm and unobstructed view and the water features hardly moved, a low distribution from −1.61 to −2.73 was obtained. The eventfulness values at sites G and H among the green spaces were −0.27 and −0.25, respectively, which were higher than the values for the waterfront areas. Although site I was a green space, it had low eventfulness. This phenomenon could be attributed to the relatively dynamic water features, such as the stream and valley water, in the green space. Regarding the overall quality of the soundscape, the values of the overall impression and appropriateness were considerably higher for the natural environments than for urban environments (see Figure 4c).

#### 3.2.2. Restoration Responses Obtained Through Soundscape Experience

Figure 5 depicts the psychological restoration responses to the soundscapes. The PRSS indicates the overall mean values of the six domains (fascination, being-away-to, being-away-from, compatibility, extent (coherence), and extent (scope)), and the SAM results are categorized as valence and arousal. The PRSS results were 2.90–3.13 for the urban environments and 4.36–4.86 for the natural environments. According to the median of the PRSS, that is, 4, the subjects produced significant psychological restoration responses to the natural environments. Additionally, the valence and arousal responses exhibited similar tendencies to pleasantness and eventfulness, respectively. The valences were high (at least 6.38) for the natural environments and relatively low (≤5.00) for the urban environments. The arousal was distributed from 4.27 to 4.68 for the urban environments, which was higher than the distribution of 2.13–2.68 obtained for the natural environments.

## 4. Discussion

### 4.1. Comparison Between Questionnaire and Physiological Soundscape Evaluation Approaches

A soundscape evaluation approach based on the physiological responses obtained through the proposed visual attention quantification technique and the questionnaire-based response results was developed. The model was intended to verify the usefulness of the physiological response-based visual attention quantification technique by examining how well the visual attention quantification technique complements conventional questionnaires. Multiple linear regression analyses were performed by setting the sound source identification results and the visual attention responses as independent variables and the perceived affective quality and overall quality of the soundscape as dependent variables. Cases in which the variance inflation factor exceeded 10 corresponded to multicollinearity and were excluded from the variables. The results obtained through this process are presented in Table 3.

Regarding the visual attention-based model results based on the responses to the questionnaires, the pleasantness was negatively affected by other noise, positively affected by the natural sounds and human sounds in the auditory sense, and positively affected by the sky in the visual sense. The eventfulness was mostly increased by the other noise and human sounds and increased by vehicles and people, indicating that this characteristic was closely associated with the occurrence of events in space. The overall impression and appropriateness exhibited similar tendencies. The overall quality increased with the increasing perception of natural elements—such as natural sounds, vegetation, and sky—and decreasing perception of negative factors, such as other noises.

The results obtained from the proposed physiological response-based visual attention evaluation model were as follows. The relationships involving the sound sources were consistent with the results from the model based on the visual attention responses obtained using questionnaires. However, the relationships involving the visual elements varied considerably. For example, in contrast to the questionnaire responses, in which the vegetation significantly affected the eventfulness, overall impression, and appropriateness, the physiological responses did not indicate a significant effect from the vegetation. Furthermore, the visual effects of the sky indicated by the questionnaires were insignificant in the physiological responses. Moreover, some of the physiological responses were not obtained through questionnaires. In particular, the positive contribution of visual attention to the water space to the eventfulness was found only in the physiological responses. A previous study [52] concluded that water features increase eventfulness, and thus, the aforementioned physiological response-based interpretation matches those results better than the questionnaire response-based interpretation.

Notable results were obtained by dividing the visual attention responses based on the number of fixations and the time duration of fixation. Based on the eventfulness model, the frequency by which the subjects gazed at the vehicles was more important than the time duration of the visual attention to the vehicles, indicating that the visual attention was affected by the movement of the vehicles changing in a short period. Thus, the eventfulness increased more significantly in a situation in which traffic moved smoothly than in a situation of congestion. Similarly, the number of fixations affected visual attention more significantly than the duration of fixation did on people. By contrast, for the water space, the duration of fixation affected the visual attention more significantly than the number of fixations did. To increase the eventfulness of the soundscape in a space, visual elements, such as vehicles and people, should move continuously instead of stopping in one space [34]. Furthermore, an environment where city dwellers can observe the movement and flow of water must be provided. By contrast, the time duration significantly affected the visual attention responses to the vegetation and sky, as indicated by the results of the pleasantness model. Therefore, green spaces and an unobstructed view of the sky should be provided to increase the pleasantness of a soundscape.

Multiple linear regression analyses were performed as described previously to examine the effects of the visual attention responses and responses to the sound sources on psychological restoration. Table 4 presents the results. The questionnaire response-based model results revealed that in the sound aspect, the natural sounds had the most positive effect on psychological restoration, whereas, in the visual aspect, the restoration responses increased with an increasing amount of visible sky and a decreasing number of buildings within a given site. By contrast, when using the model based on the visual attention responses quantified by the eye-tracking technology, the negative effect of the buildings and the positive effect of the sky were similarly verified, and the negative effects of vehicles and people were additionally verified. As in the eventful responses described earlier, in the case of moving objects such as vehicles and people, statistical significance was clearly exhibited by the number of fixations. In particular, the vehicles significantly affected the time duration of fixation, indicating that the absolute quantity of vehicles within a space must be controlled. In this study, multiple linear regression analysis was performed to calculate the adjusted R^2^ to compare the explanatory power differences between the survey-based soundscape evaluation approach and the physiological measurement-based evaluation approach. The adjusted coefficient of determination is an indicator of explanatory power, representing the fit of the data and adjusting for bias according to the number of variables. Therefore, evaluating how well each statistical analysis result can explain the variability in soundscape perception is possible. The interpretation results of the psychological restoration of soundscapes revealed that the adjusted coefficients of the survey-based model ranged from 0.27 to 0.41 (mean = 0.33, standard deviation = 0.07), whereas the physiological measurement-based model exhibited an explanatory power from 0.28 to 0.43 (mean = 0.35, standard deviation = 0.08). In particular, the Valence results showed a maximum explanatory power difference of 2%. Through this, physiological data can provide more intuitive and objective information in interpreting cognitive responses to soundscapes, increasing the accuracy of the evaluation. In particular, eye tracking can complement intentional and subjective evaluations through surveys by capturing humans’ unconscious reactions to visual stimuli.

Finally, to achieve a comprehensive understanding of soundscape, this study conducted a multiple linear regression analysis using both subjective response and physiological response data, and the results are presented in Table 5. Therefore, although the existing two statistical analysis results (Table 3 and Table 4) exhibited similar trends, the overall statistical analysis approach demonstrated higher explanatory power than the adjusted R^2^ value. For example, in the case of arousal, when only the survey was used, the value was 0.27, but when combined data were used, the value exhibited a higher explanatory power of 0.29. Thus, this result reaffirms that physiological responses can complement existing survey-based research methods. In particular, when the questionnaire measures the amount of visual element recognition, by using physiological responses together, it can extend the interpretation of the perception of various elements that constitute the context of the space into a multidimensional analysis of gaze frequency and duration.

### 4.2. Limitations and Further Study

This study had the following limitations. First, only fixation and time duration were examined as quantification indices for eye movement. In future studies, other indices—such as saccades or scan paths—should be considered. Furthermore, this study used a single value of 300 ms for the threshold of fixation, whereas previous studies used various thresholds, such as 100 or 200 ms [53]. Furthermore, because of the possibility that eye movement can be controlled by the subject’s subjectivity during the experiment, attention or disregard for specific elements may occur. This subjective control can hinder the consistency of visual responses and affect the interpretation of research results. Therefore, devising methodologies in future research that can exclude these influencing factors is crucial. Thus, the effect of varying the threshold value should be examined. Moreover, in this study, the static factors were distinguished manually using Adobe Photoshop. Therefore, methods to reduce the burden on researchers using deep-learning-based image segmentation techniques should be considered. Furthermore, a physical analysis of the visual environment was performed in this study. However, if further research quantifies the proportion of space occupied by green areas or the sky and compares it with the results of this study, then it will demonstrate the reliability of the proposed methodology in quantifying human perceptual responses to the visual environment. Another limitation was the homogeneity in the demographic of the participants recruited for this study, who were Korean students in their 20s, resulting in limited generalizability. Generally, older adults have a slower processing speed for visual information and a reduced ability to perceive information in their peripheral vision [54]. Younger people respond quickly to visual stimuli, tend to have longer saccades, and explore a wider field of vision. By contrast, older adults tend to gaze at visual stimuli for longer and reduce cognitive load through repetitive fixation. These age-related differences are evident in the eye-tracking results, revealing variations in gaze frequency and duration [55]. Therefore, to derive generalized research results on the relationship between soundscape perception and visual attention patterns, conducting studies targeting various age groups is crucial. Furthermore, studies involving countries with various social, cultural, and architectural contexts are necessary. Furthermore, because the effect of familiarity with space on landscape and soundscape perception or preference is evident, investigating the extent to which familiarity affects physiological responses such as eye tracking is crucial [56,57]. Additionally, the eye-tracking accuracy should be verified in cases of eyesight degradation or extraocular muscle capability degradation caused by aging. In this study, downmixing headphones were used to provide stereophonic sound. However, additional findings can be obtained if more realistic ecological validity with advanced reproduction technology is provided. Because this study was laboratory-based, verification through comparison with actual field response results is essential. However, despite these limitations, this study is crucial in that it provides a method for quantifying eye movement among various physiological measurements in the soundscape field and is scientifically meaningful in that the proposed objective measurement method complements conventional questionnaires. However, because soundscape perception is a complex mechanism depending on contextual environmental conditions, several limitations exist in evaluating the proposed method, which is based solely on eye-movement quantification. The proposed method can be used as a supplementary tool to be used in conjunction with existing soundscape evaluation methodologies.

## 5. Conclusions

In this study, a novel methodology was proposed for examining the effect of visual elements on soundscape evaluation based on objective eye-tracking measurements. In contrast to previous studies using eye-tracking technology, eye tracking was performed for both dynamic and static elements of the spatial environment by using a deep-learning algorithm. The results of the proposed physiological measurement-based objective evaluation methodology were consistent with those of the conventional questionnaire-based subjective evaluation methodology. The negative effects of artificial elements (vehicles) and the positive effects of natural elements (vegetation, water features, and sky) on soundscape perception were similar for both evaluation methods. We observed that the soundscape restoration response based on physiological measurement had a 1–2% higher explanatory power than that of the questionnaire-based model. Therefore, questionnaires intended to examine visual attention responses to visual elements can be complemented with measurements to reduce the fatigue of subjects and achieve an objective evaluation. In particular, causal relationships among the time-series characteristics of visual elements can be interpreted according to soundscape quality and restoration responses. The proposed method is an effective method of examining visual attention through physiological responses and can be considered an alternative or supplementary tool for surveying methods that can be affected by the subjectivity of researchers. However, although this study is significant in its ability to investigate the physiological response of vision to soundscape perception, technical limitations to eye tracking remain. Furthermore, other contexts can reveal insights into visual elements not covered in this study. Therefore, the proposed methodology should be validated in various contexts. The process of quantifying eye tracking in this study is valuable in that soundscape researchers can easily reproduce it. The proposed method can be used to efficiently examine the effects of visual environments on soundscape assessment through simple measurements and can support the design and planning of urban soundscapes.

## Figures and Tables

**Figure 3 ijerph-21-01478-f003:**
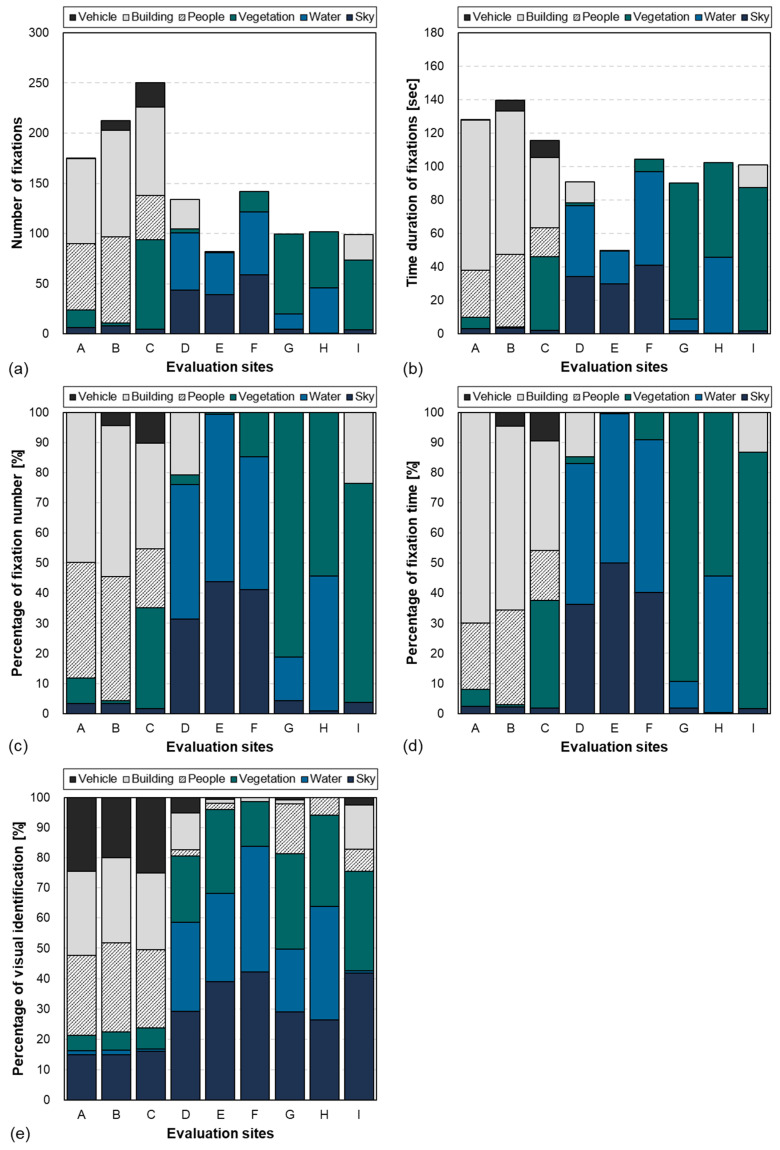
Visual attention responses obtained using eye-tracking technology: (**a**) number of fixations; (**b**) duration of fixations; (**c**) percentage of fixation number; (**d**) percentage of the fixation time; and (**e**) percentages of visual dominance based on questionnaire responses.

**Figure 4 ijerph-21-01478-f004:**
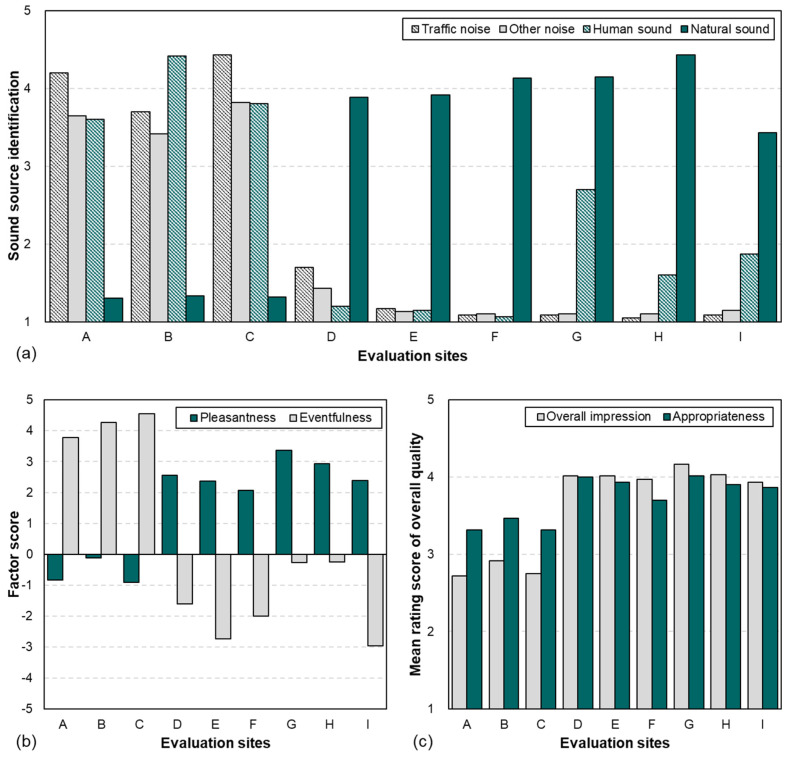
Soundscape perception results: (**a**) sound source identification; (**b**) perceived affective quality; and (**c**) overall soundscape quality.

**Figure 5 ijerph-21-01478-f005:**
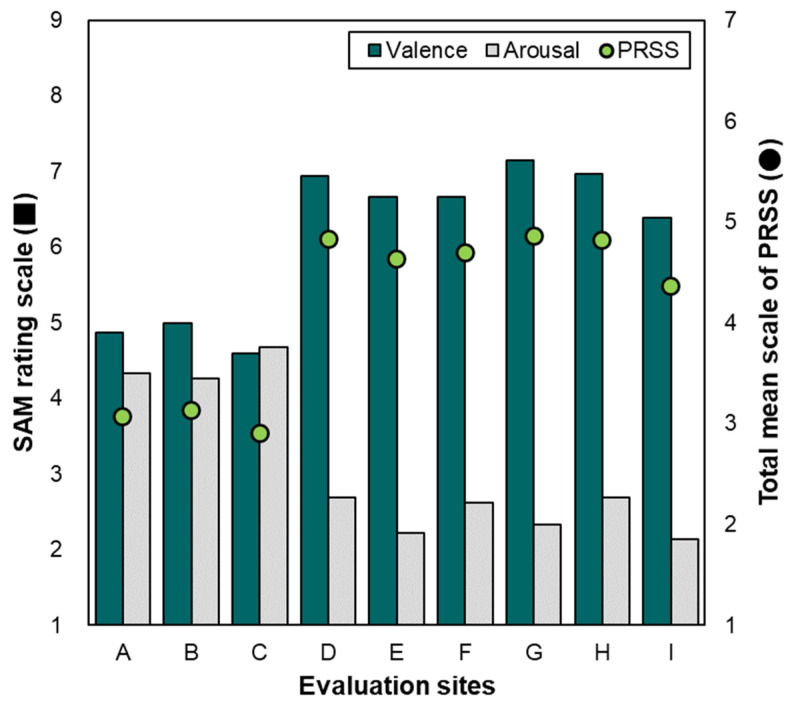
Psychological restoration responses for various soundscape experiences.

**Table 1 ijerph-21-01478-t001:** Visual and acoustic characteristics of the nine evaluation sites.

Parameter	Visual Parameter [%]				Acoustic Parameter [dB]		
Site	Green	Water	Sky	Grey	L_Aeq_	L_Ceq-Aeq_	L_A5-A95_
Urban							
(a) High-density commercial area	7	0	18	73	75.9	12.4	16.5
(b) Low-density commercial area	3	0	25	70	70.6	11.5	17.4
(c) Business area	0	0	18	62	74.2	12.9	20.6
Waterfront							
(d) River	8	8	45	0	50.7	27.2	10.3
(e) Wetland	10	1	47	0	45.4	16.4	7.1
(f) Ocean	0	0	47	0	44.3	12.7	5.2
Green space							
(g) Forest	40	2	7	0	48.9	10.1	11.6
(h) Valley	41	4	4	0	67.2	6.3	10.0
(i) Temple	37	0	10	3	51.7	16.6	21.5

**Table 2 ijerph-21-01478-t002:** Pearson correlation between physical characteristics and visual response.

Parameter	Visual Parameter [%]				Acoustic Parameter [dB]		
Site	Green	Water	Sky	Grey	L_Aeq_	L_Ceq-Aeq_	L_A5-A95_
Number of fixations [num]							
Vehicle	−0.18 **	−0.34 **	−0.11 **	0.53 **	0.49 **	−0.11 **	0.47 **
Building	−0.40 **	−0.46 **	−0.12 **	0.85 **	0.67 **	−0.28 **	0.46 **
People	−0.43 **	−0.51 **	−0.13 **	0.89 **	0.73 **	−0.21 **	0.50 **
Vegetation	0.59 **	−0.25 **	−0.61 **	−0.05	−0.19 **	0.49 **	−0.14 **
Water space	−0.14 **	0.71 **	0.50 **	−0.56 **	−0.40 **	−0.05	−0.04
Sky	−0.47 **	0.52 **	0.71 **	−0.33 **	−0.12 **	−0.22 **	0.23 **
Time duration of fixations [s]							
Vehicle	−0.23 **	−0.37 **	−0.11 *	0.59 **	0.52 **	−0.13 **	0.49 **
Building	−0.41 **	−0.48 **	−0.13 **	0.86 **	0.68 **	−0.25 **	0.42 **
People	−0.43 **	−0.49 **	−0.11 **	0.85 **	0.69 **	−0.21 **	0.47 **
Vegetation	0.77 **	−0.21 **	−0.66 **	−0.26 **	−0.43 **	0.52 **	−0.45 **
Water space	−0.09 *	0.67 **	0.35 **	−0.47 **	−0.41 **	0.02	0.01
Sky	−0.41 **	0.50 **	0.66 **	−0.33 **	−0.12 **	−0.23 **	0.17 **
Subjective response							
Vehicle	−0.34 **	−0.45 **	−0.15 **	0.81 **	0.70 **	−0.21 **	0.51 **
Building	−0.41 **	−0.46 **	−0.12 **	0.87 **	0.71 **	−0.27 **	0.51 **
People	−0.23 **	−0.56 **	−0.32 **	0.85 **	0.68 **	−0.04	0.45 **
Vegetation	0.37 **	0.12 **	−0.09 *	−0.42 **	−0.34 **	0.14 **	−0.42 **
Water space	−0.05	0.74 **	0.44 **	−0.63 **	−0.47 **	0.03	−0.10 *
Sky	−0.14 **	0.39 **	0.46 **	−0.45 **	−0.28 **	−0.12 **	−0.05

* *p*-value < 0.05; ** *p*-value < 0.01.

**Table 3 ijerph-21-01478-t003:** Relationship between the perception of audio–visual elements and soundscape quality.

	Physiological Response								Subjective Response			
Methodology	Eye Tracking								Questionnaire			
Item	Number of Fixations [num]				Time Duration of Fixations [s]				Identification			
Adjusted R^2^	0.41	0.53	0.38	0.12	0.41	0.53	0.37	0.11	0.42	0.51	0.39	0.14
Dimension	PL	EV	OI	AP	PL	EV	OI	AP	PL	EV	OI	AP
Traffic noise	−0.20 *	−0.35 **	−0.20 *	−0.15	−0.17	0.37 **	−0.17	−0.11	-	-	-	-
Other noise	−0.31 **	0.03	−0.15	−0.06	−0.30 **	0.04	−0.16	−0.05	−0.35 **	0.14 **	−0.26 **	−0.05
Human sound	0.21 **	0.28 **	0.09	0.19 **	0.21 **	0.28 **	0.08	0.17 *	0.19 **	0.29 **	0.04	0.15
Natural sound	0.35 **	0.03	0.32 **	0.32 **	0.32 **	0.05	0.29 **	0.28 **	0.31 **	0.04	0.29 **	0.24 **
Vehicle	−0.03	0.10 *	−0.02	−0.02	−0.03	0.07	−0.04	−0.02	−0.08	0.17 *	0.01	−0.16
Building	0.13	−0.12	0.08	0.19	0.05	0.04	−0.08	0.02	−0.01	0.01	0.09	0.03
People	−0.11	0.24 **	−0.16	−0.16	−0.01	0.13 *	−0.02	0.01	−0.02	0.18 **	0.01	0.01
Vegetation	−0.01	0.03	−0.05	−0.04	0.09 *	0.06	0.02	0.08	0.04	−0.10 **	0.10 **	0.10 *
Water space	−0.02	0.10 *	−0.04	−0.06	0.05	0.13 **	−0.01	0.01	−0.09	0.02	−0.06	−0.10
Sky	0.03	−0.09 *	0.06	0.07	0.08 *	−0.06	0.05	0.09	0.14 **	−0.04	0.14 **	0.18 **

PL: pleasantness; EV: eventfulness; OI: overall impression; AP: appropriateness; * *p*-value < 0.05; ** *p*-value < 0.01.

**Table 4 ijerph-21-01478-t004:** Relationship between perception of audio–visual elements and psychological restoration.

	Physiological Response						Subjective Response		
Methodology	Eye Tracking						Questionnaire		
Item	Number of Fixations [num]			Time Duration of Fixations [s]			Identification		
Adjusted R^2^	0.42	0.33	0.28	0.43	0.33	0.28	0.42	0.31	0.27
Dimension	PRSS	Valence	Arousal	PRSS	Valence	Arousal	PRSS	Valence	Arousal
Traffic noise	−0.10	0.09	0.10	−0.08	0.10	0.12	-	-	-
Other noise	−0.04	−0.37 **	0.26 **	−0.05	−0.37 **	0.25 **	−0.05	−0.28 **	0.27 **
Human sound	0.09	0.18 **	−0.03	0.10	0.17 **	−0.01	0.04	0.14 *	0.01
Natural sound	0.40 **	0.39 **	−0.08	0.36 **	0.36 **	−0.07	0.36 **	0.33 **	−0.09
Vehicle	−0.13 **	−0.08	0.06	−0.13 **	−0.09 *	0.06	−0.03	−0.01	0.01
Building	0.06	0.05	−0.10	−0.13 *	0.01	−0.10	−0.19 *	−0.05	0.17 *
People	−0.18 *	−0.05	0.27 **	−0.02	0.01	0.19 **	−0.03	0.01	0.05
Vegetation	0.03	0.03	0.05	0.04	0.06	−0.05	0.05	0.04	−0.03
Water space	−0.10	−0.04	0.13 *	0.04	−0.03	0.10 *	0.01	−0.03	0.13 *
Sky	0.14 **	0.17 **	−0.08	0.14 **	0.17 **	−0.09 *	0.16 **	0.09 *	−0.09 *

* *p*-value < 0.05; ** *p*-value < 0.01.

**Table 5 ijerph-21-01478-t005:** Holistic soundscape evaluation approaches with subjective and physiological responses.

Dimension	Soundscape Quality				Psychological Response		
	Pleasantness	Eventfulness	Overall Impression	Appropriateness	PRSS	Valence	Arousal
Adjusted R^2^	0.43	0.53	0.40	0.14	0.42	0.33	0.29
Subjective response: sound perception							
Traffic noise	-	-	-	-	-	-	-
Other noise	−0.32 **	0.12	−0.23 **	−0.02	−0.23 **	−0.28 **	0.23 **
Human sound	0.19 **	0.26 **	0.04	0.15	0.12	0.18 *	0.00
Natural sound	0.30 **	0.06	0.28 **	0.24 **	0.32 **	0.33 **	−0.06
Subjective response: visual perception							
Vehicle	−0.08	0.20 **	0.01	−0.17	0.02	0.01	−0.02
Building	−0.13	0.00	−0.08	−0.01	−0.07	−0.10	0.21 *
People	0.04	0.11	0.06	0.04	0.02	0.03	−0.01
Vegetation	0.05	−0.08 *	0.10 *	0.09 *	0.07	0.06	−0.01
Water space	−0.09	−0.01	−0.07	−0.08	−0.09	−0.07	0.10
Sky	0.15 **	0.00	0.11 **	0.16 **	0.11 *	0.06	−0.06
Physiological response: Number of fixations [num]							
Vehicle	-	-	-	-	-	-	-
Building	0.17	−0.17 *	0.17	0.25 *	0.17	0.10	−0.07
People	-	-	-	-	-	-	-
Vegetation	−0.10	0.06	−0.18 **	−0.16 *	−0.12 *	−0.03	0.18 **
Water space	−0.05	0.02	−0.04	−0.06	0.01	0.04	−0.02
Sky	−0.04	−0.06	0.11	0.08	0.02	0.07	−0.04
Physiological response: Time duration of fixations [s]							
Vehicle	−0.04	0.09 *	−0.04	−0.03	−0.11 *	−0.09	0.03
Building	−0.02	0.14 *	−0.16 *	−0.10	−0.09	0.00	−0.08
People	−0.06	0.17 *	−0.07	−0.07	−0.12	−0.02	0.22 **
Vegetation	0.16 **	0.01	0.15 *	0.20 **	0.10	0.07	−0.18 **
Water space	0.11	0.12 *	0.05	0.07	0.03	−0.04	0.09
Sky	0.10	0.00	−0.05	0.00	0.12	0.12	−0.06

* *p*-value < 0.05; ** *p*-value < 0.01.

## Data Availability

The data that support the findings of this study are available on request from the corresponding author.

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
