# Peer review of "Quantification of Visual Attention by Using Eye-Tracking Technology for Soundscape Assessment Through Physiological Response"

_ijerph, 2024, doi:10.3390/ijerph21111478_

Round 1
Reviewer 1 Report
Comments and Suggestions for Authors
This manuscript provides a new method for evaluating the physiological response of soundscapes, and analyzes the results of eye-tracking technology with those of traditional questionnaire surveys. The topic is meaningful, the content is novel, and the conclusions are valuable. There are two problems that can be supplemented.
1. Does the familiarity of participants with the 9 evaluation points affect the measurement results?
2. It is necessary to supplement the correlation analysis between the measurement results of eye-tracking technology and the physical characteristic indices of 9 evaluation points, in order to analyze the impact of objective environmental differences on eye tracking technology data.
Reviewer 2 Report
Comments and Suggestions for Authors
Title: Quantification of visual attention using eye-tracking technology for soundscape assessment through physiological response
1. Overview and general recommendation
The idea of a framework to research on using eye-tracking technology for soundscape assessment is interesting. However, I think that the descriptions of some very important points were inadequate. I recommend that a major revision is warranted. I explain my concerns in more detail below. I ask that the authors specifically address each of my comments in their response.
2. Major comments
1) For the Keywords, the descriptions of “soundscape” and “perceptions” were so broad and it is recommended much more focused ones.
2) For 2.2.1 section, how can the high density and low density of point A and point B be distinguished from the visual perspective? Are there differences and representativeness in the visual images of fixed points?
3) It is recommended to explain the reliability of the results obtained in this laboratory experiment instead of field experiment.
4) From Line 225 to Line228, what were the degrees of visual perception? What were the detailed descriptions of visual element identification?
5) The respondents were all young people. Further discussion is suggested in the Discussions sections.
6) In conclusions at Line 507, how does the 1-2% result reflect in the previous results?
3. Minor comments
1) There was no reference for ISO 12913-2 at Line 223.
2) Are positive and negative scales in the same direction for all scales? Further explanations are suggested in 2.3.2 Questionnaires.
3) Usually, the figures came after the text for Figure 3 and 4. It is recommended to adjust the order of text and figures for Figures 3 and 4.
Reviewer 3 Report
Comments and Suggestions for Authors
The article presents a methodology for investigating the impact of visual elements on soundscape evaluations using objective eye-tracking measurements, based on data from 498 participants. The article makes a significant contribution, and the results are clearly presented. However, some adjustments are needed, as outlined below:
L73 – Could you provide an example of an urban planning decision focused on soundscapes, ideally from the authors' country?
Figure 2 – Please clarify why you chose a 3-minute duration for the 3D audio-visual environment experience.
L214 – What about the hearing conditions of your sample? Did you test their hearing thresholds? If not, you should explain this omission in the article.
Study limitation: You should address the fact that eye movements can be subjectively controlled, as mentioned in line 263.
I recommend a minor revision.
Reviewer 4 Report
Comments and Suggestions for Authors
-
Page 3, Line 143: The use of the YOLO tool should be validated for accuracy in the context of this study. While this tool is powerful, it may not necessarily be accurate within the scope of this research.
-
Page 13, Line 39: The study explores the correlation between eye-tracking data results and questionnaire results with soundscape perception. However, there is a lack of principle-based analysis and mathematical modeling based on principles, making it difficult to be referred to as a "soundscape perception model." A change in phrasing is recommended.
-
Page 14, Line 45: The eye-tracking data results interpret human visual perception from a process perspective, while subjective evaluation questionnaires provide the outcomes of perception. Theoretically, combining these two could yield better results. The authors are encouraged to conduct further analysis of the data.
-
Page 15, Line 507: "We observed that the soundscape perception model based on physiological measurement had a 1–2% higher explanatory power than that of the questionnaire-based model." How was this conclusion drawn? It was not demonstrated in the results section.
Round 2
Reviewer 2 Report
Comments and Suggestions for Authors
The idea of using eye-tracking technology for soundscape assessment is interesting. The authors answered the questions well and made changes in the manuscript. I recommend an accept for the revised version for this manuscript.